# Long-Term Follow Up of Sexual Function and Steroid Levels in Women after Perimenopausal Hysterectomy with or without Concomitant Oophorectomy

**DOI:** 10.3390/jcm12154976

**Published:** 2023-07-28

**Authors:** Jonas Zimmerman, Mats Brännström, Cornelia Bergdahl, Adel Aziz, Jonas Hermansson

**Affiliations:** 1Department of Obstetrics and Gynecology, Institute of Clinical Sciences, Sahlgrenska Academy, University of Gothenburg, 405 30 Göteborg, Swedenadel.aziz@vgregion.se (A.A.); 2Department of Research and Development, SV Hospital Group, 424 22 Angered, Sweden

**Keywords:** sexuality, testosterone, hysterectomy, oophorectomy, ovary

## Abstract

Hysterectomy, most often performed because of bleeding disorders or uterine leiomyoma, is one of the most common major surgical procedures in women and is usually performed during the perimenopausal period on ages 45–55 years. Hysterectomy may be combined with bilateral salpingo-oophorectomy, as a risk-reducing procedure to minimize the risk of ovarian cancer. An open question is whether concomitant oophorectomy, with cessation of ovarian androgen secretion, has any long-term effects on sexual function. In the present prospective cohort study of women undergoing benign hysterectomy, the long-term (10–12 years) effects on sexual function and changes in sex hormone levels were investigated in women having undergone perimenopausal hysterectomy, with or without concomitant bilateral salpingo-oophorectomy. Originally, 491 women (mean age around 50 years) were operated with (patient preference) either only hysterectomy (HYST) or hysterectomy plus bilateral salpingo-oophorectomy (HYST + BSO), and 441 women (90%; HYST; *n* = 271 and HYST + BSO; *n* = 170) completed a one-year survey. In the present study, 185 women (42%) of the cohort with one-year follow up participated in the long-term follow up after 10–12 years. Follow-up was with the 10-item McCoy Female Sex Questionnaire and blood analysis of levels of testosterone, estradiol and sexual-hormone-binding globulin. The results showed that specific aspects of sexual function were lower after HYST + BSO compared to HYST 10–12 years after surgery. These lowered items were frequency of sexual fantasies, enjoyment of sexual activity, sexual arousal, and orgasmic frequency. No long-term differences in sex hormone levels were found between the two groups. In conclusion, some items related to sexual function were lower after HYST + BSO in a long-term perspective study, although the levels of testosterone were unaltered. This finding may have implications for clinical recommendations concerning prophylactic salpingo-oophorectomy or for hysterectomy during the perimenopausal age.

## 1. Introduction

Hysterectomy is the second most common major surgical procedure of women, with more than 430,000 hysterectomies performed in the USA in 2010 [1]. More than 90% of hysterectomies are conducted on benign indications, with leiomyoma, bleeding and endometriosis representing the primary diagnosis of around 30%, 22% and 16%, respectively [2]. More than two-thirds of all hysterectomies are performed before menopause, with the highest incidence in the age span 40–49 years (43%) [1].

The uterus does not exert any endocrine function. Surgical removal of the uterus is a reasonable definitive option for women experiencing benign uterine diseases impacting quality of life or resulting in chronic problems [3].

When performing hysterectomy surgery, it is possible to also perform a bilateral salpingo-oophorectomy, in order to minimize the risk of future ovarian malignancy. However, the number needed to treat (NNT) with bilateral salpingo-oophorectomy to avoid one case of ovarian cancer is shown to be somewhere between 273 and 323 [4,5]. Solid data show that premenopausal oophorectomy should only be performed if strictly necessary, and along with substitution with estrogens until the age of natural menopause [6,7]. There are also studies showing that oophorectomy in conjunction with hysterectomy is likely to be associated with increased morbidity and mortality [8,9].

There are controversies regarding how oophorectomy affects sexual function expressed as libido, sexual arousal, orgasmic frequency, and satisfaction. Some studies suggested that the rapid androgen decline associated with oophorectomy would negatively affect sexual function [10,11], at least in the short-term perspective. However, a Swedish prospective one-year study, including the patient material of the present study, assessing the androgen levels and sexual function one year after hysterectomy with or without oophorectomy could see only a slight decline in levels of ovarian-derived androgens after oophorectomy and no difference in sexual function when comparing the two groups one year after surgical intervention [12]. The prevalence of hysterectomy and the potential effects of the surgery on aspects of sexual function calls for further research of this as a part of increasing the understanding of health in aging women. The aim of the present study was to extend the Swedish study material mentioned above and compare the long-term (10–12 years) effects on sexual function after only hysterectomy with the combined procedure of hysterectomy and bilateral salpingo-oophorectomy. 

## 2. Material and Methods

### 2.1. Study Sample

The cohort for the present study incorporates more than 300 women who were hysterectomized with or without concomitant bilateral salpingo-oophorectomy and followed prospectively to assess potential effects on sexual function and androgen levels. Unilateral salpingo-oophorectomy was performed because ovarian cysts were discovered during surgery, and malignancy could not be ruled out on 12% of the women in the HYST group; histologic evaluation proved these adnexal cysts as benign. Subtotal hysterectomy was more common in the HYST group (24%) than in the HYST + BSO group (4%). No differences between the two groups regarding educational level, occupational status, tobacco smoking, previous use of oral contraceptives, gynecologic surgical intervention, age at menarche, or number of pregnancies or deliveries was present. The preoperative body mass index was similar in the two groups and remained stable during the post-operative follow-up. At the short-term follow-up, no negative effects regarding sexual function were seen in either group, these results are described in detail in a Ph.D thesis from 2004 [12].

### 2.2. Patients

The study sample included women scheduled for elective hysterectomy on benign indication. The study was approved by the Gothenburg Regional Ethics Review Board (reference no.: 506-07) and all women gave their informed consent to participate. Initial recruitment took place between March 1996 and December 1999. Women of perimenopausal age (45–55 years) scheduled for elective hysterectomy at Sahlgrenska University Hospital in Gothenburg, Sweden, or at the associated regional Borås Hospital, Sweden, were invited to participate. Inclusion criteria were being part of a partner relationship, an age of 45–55 years, last menstruation ≥12 months prior, scheduled for hysterectomy on benign indication, and being sexually active. For this study, being sexually active was defined as ≥one episode of intercourse per month during the past 6 months before the start of the study. Exclusion criteria were psychiatric disease that could interfere with the studied outcome, e.g., depression, and previously seeking medical care for sexual problems.

In total, 491 women were included in the study and 441 completed the short-term (one year) follow-up while 50 women were lost during follow-up. The gynecologist informed the women their option to either undergo only a hysterectomy (HYST group) or to undergo a hysterectomy with a salpingo-oophorectomy (HYST + BSO group). Thus, this patient preference-selection study included 271 women in the HYST group and 170 in the HYST + BSO group. A total of 32 women in the HYST group underwent unilateral salpingo-oophorectomy since ovarian cysts were discovered during surgery, hence malignancy could not be dismissed, and they were excluded from further analysis in this study. Ten to twelve years after surgery, a long-term follow-up study on the same cohort was initiated. Of the original 441 women, 185 (41.9%) completed the long-term follow-up questionnaires, 115 (62.2%) in the HYST group and 70 (37.8%) in the HYST + BSO group. The comparisons in this paper are on differences between the groups in the long-term follow-up and not on differences between each woman at short- and long-term follow-up.

## 3. Procedures

At inclusion, the women met the surgeon who would perform the surgery. The woman was informed considering the possibility of also performing a prophylactic bilateral salpingo-oophorectomy at the same time as the hysterectomy and thereafter selected one of the two procedures. At inclusion and one year after surgery, blood samples for the analysis of hormones were taken and the women were asked to fill in the 10-item Swedish version of the McCoy’s Female Sex Scale (MFSQ). Data from baseline (before surgery) and at one year after surgery have been published [12].

The present study is a follow-up using data collected 10–12 years after surgery, including the women of the HYST group and the HYST + BSO group. The women were invited by mail to participate, and one co-author (CB) saw all the women for a gynecological examination. Blood samples were taken for analysis (see below) of the same hormones as at baseline and at the one-year observation. The women were asked to fill in the MFSQ at home, to be returned by mail to the researchers. 

### Assays

Venous sex blood hormones were taken preoperatively, at one-year follow-up and after 10–12 years. The samples of venous sex blood hormones were taken during office hours, the samples were not standardized to avoid menstrual cyclical or circadian variations. The blood samples were immediately centrifuged, and the plasma was stored at 20 °C. Plasma concentrations of testosterone (T) (nmol/L), estradiol (E) (nmol/L), and sex-hormone-binding globulin (SHBG) (nmol/L) were determined by time-resolved fluoro immunoassay using DELFIA® kits obtained from Wallac Oy, Turku, Finland. The free androgen index (FAI) was calculated from the ratio 100 × (T/SHBG). Of the 185 women completing the long-term follow-up questionnaires, 77 had complete blood samples taken. 

## 4. Questionnaire 

Sexual function was assessed by MFSQ, which originally contained 19 items [13]. This was later been modified to a 10-item version that has previously been used in studies investigating sexual functioning in perimenopausal women [14,15]. This study used the Swedish translation version of MFSQ. This scale is focused on sexual experience and responsiveness during the last 30 days. Each item addresses different aspects of sexual life as follows: 1. satisfaction with present sexual activity; 2. frequency of sexual fantasies; 3. enjoyment of sexual activity; 4. sexual arousal; 5. orgasmic frequency; 6. lubrication; 7. dyspareunia; 8. relationship to the partner as a lover; 9. relationship to the partner as a friend; and 10. coital frequency. The MFSQ scale consists of seven grades where a high value signifies high satisfaction and a low value low satisfaction.

### Statistical Methods

For the included variables, the differences between the two types of surgery were analyzed with Mann–Whitney U tests or independent sample *t*-tests depending on the type of data. A *p*-value of 0.05 or less was considered statistically significant. All calculations were performed using SPSS v21 for PC. 

## 5. Results

The HYST group members were on average 3 years younger than the HYST + BSO group (Table 1). Mean weight, mean age at menarche, number of pregnancies and deliveries were similar in both groups.

Table 2 shows differences in hormone levels between the groups and longitudinal changes within the groups between short-term and long-term follow-up. The hormone levels were similar in both the HYST group and HYST + BSO group at the short-term follow-up for testosterone, estradiol, and the free androgen index. Between the one year and 10–12 years follow-ups, these levels declined in both groups. SHBG was higher in the HYST group at the short-term follow-up. The SHGB value decreased slightly in the HYST group between follow-ups. This was not the case for the HYST + BSO group where the SHBG levels increased between follow-ups. However, these changes were non-significant. No other differences considering sex hormone levels were found between the groups. 

Table 3 shows rank mean values on MSFQ questionnaires between HYST and HYST + BSO at the long-term follow-up. The HYST + BSO group had lower values on frequency of sexual fantasies, enjoyment of sexual activity, sexual arousal, and orgasmic frequency. No other differences were found.

## 6. Discussion

The present study is a unique, more-than-10-year follow up of the sexual function after hysterectomy with or without concomitant bilateral salpingo-oophorectomy. No other long-term prospective studies of these or similar patient groups and with sexuality and hormone levels as endpoints exist. The major findings were that specific aspects of the sexual function were higher in the HYST group than in the HYST + BSO group in a long-term perspective. These specific items were sexual fantasies, sexual arousal, enjoyment of sexual activity and orgasmic frequency. There were no differences between the groups when comparing satisfaction with the partner as a friend or lover, vaginal dryness, or dyspareunia.

The results of sexual function after hysterectomy with concomitant bilateral salpingo-oophorectomy are in line with earlier research. A systemic review from 2016, however, lacking long-term prospective studies over 5 years, concluded hysterectomy alone or together with unilateral oophorectomy to be less deteriorating on sexual function than hysterectomy with concomitant bilateral salpingo-oophorectomy [16]. Rhodes and coworkers [17] linked anorgasmia to bilateral salpingo-oophorectomy at a one-year follow-up. Another study found, in a five-year follow-up, that women with bilateral salpingo-oophorectomy in a higher degree suffered from loss of libido and difficulties with sexual arousal [11]. Other studies have not found this association; Teplin and coworkers [18] found at no time of their follow-up visits (6, 12, 18, 24 months) differences in sexual function between hysterectomy and hysterectomy with bilateral salpingo-oophorectomy. Kokcu and coworkers [19] reported significant worsening of vaginal dryness after hysterectomy with bilateral salpingo-oophorectomy compared to hysterectomy but no differences in overall sexual function. The population of the present study was based on material from a previous short-term follow-up that did not find any differences in sexual function between hysterectomy and hysterectomy with bilateral salpingo-oophorectomy at follow up one year after surgery [12]. Thus, it may be that potential the negative effect on sexuality by oophorectomy and loss of ovarian steroids are more apparent in a longer time frame. 

It is likely to assume that multiple factors contribute to the results on sexual function in this study. The composition of the study population, the inherent motive of one specific questionnaire assessing sexual function, the use of hormone replacement therapy (HRT) with estrogens during the initial years after oophorectomy, but not in the long run, may provide partial explanations. 

In the present study, the HYST + BSO group was slightly older compared to the HYST group and that may affect the results concerning sexuality. The same has been found in earlier studies on this issue [11]. A suggested factor has been that increasing age often indicates a decline in enjoyment and interest of sexual activities and therefore might affect the result [20]. Aziz [12] proposed a tendency of women with well-functioning sexual life to be prone to choosing hysterectomy rather than hysterectomy with bilateral salpingo-oophorectomy when considering the procedure. The reason for this was not fully understood but possible explanations were local praxis and how the information was given to the woman prior to the procedure.

The present study results concerning sexuality relied on the MSFQ questionnaire which focus on a broad range of aspects of sexual life. The questionnaire covers physiological as well as psychosexual aspects of the sexual function with questions including the satisfaction of the partner as a friend and the satisfaction of the partner as a lover. While sexual function is indeed a multi-facetted function, based on several physiological and psychological items, the possibilities of a complete assessment from a 10-item questionnaire could be questioned. Moreover, for the evaluation of the appropriate surgical method, the primary interest is to investigate physiological effects and hormonal changes of the procedure that could impact the sexual function. In recent years, the Female Sexual Function Index (FSFI) has been the most widely known assessment tool for sexual function with six key domains: desire, arousal, lubrication, orgasm, satisfaction, and pain [21]. The focus of the FSFI is to a lager extent in line with the physiological reactions to sexual stimuli and it is possible that for the present study, if it used the FSFI questionnaire, its findings may have differed. Another issue that is worth mentioning is that we do not have data on the number of women that were sexually inactive. In total, 10 women responded to have a very unsatisfactory level of current sexual activity and did not complete the rest of the MSFQ. It is likely that our interpretation that these women were sexually inactive is correct, but we cannot be sure. 

In this study, the 10-item MSFQ was used; if we would have used the full 19 items questionnaire, we would have had access to the question on frequency of sexual intercourse. However, that concerns the last four weeks and only sexual activity in the form of intercourse. It may not entirely capture relevant aspects of whether a person considers themself as sexually active or not. If we would conduct this study again, we would include a question similar to “Do you consider yourself to be sexually active?” with the following alternatives: Yes; No; No, but I would like to be; and a possibility of a free text option. By doing so, we believe that we would more accurately capture the possible dimensions of this question for aging women. The number of women that completed each question of the MSFQ varies, as shown in Table 3. We believe that this may be an indication that they found some of the questions irrelevant, which could indicate that they were sexually inactive. This means that the results are preferably interpreted per question in the MSFQ. 

The effect of HRT on sexual function after bilateral salpingo-oophorectomy is debated. Adelman and Sharp [10] suggest in a large review that HRT might lower the negative consequences on lubrication and vaginal dryness after bilateral salpingo-oophorectomy but on the aspects of sexual arousal and overall enjoyment of sexual activities, the impact is yet to be explained. In this study, there was not enough data on the use of HRT, and we were not able to draw further conclusions, but this is a factor that would be beneficial to address in future studies on the subject. However, considering that Swedish recommendations are avoid systemic HRT after the age of 60 years for menopausal symptoms, it is likely that very few women of either group used systemic estrogens [22]. Moreover, there were no differences in estradiol levels between the two groups, further indicating that the use of exogenous estrogens was not different between the two groups. 

Concerning levels of testosterone, estradiol, and the free androgen index there were not different between the HYST and HYST + BSO group at the follow-ups. At the short-term follow-up, the HYST group had higher levels of SHGB compared to the HYST + BSO group. Between follow-ups the SHBG for the HYST group decreased while in the HYST + BSO group, the change was an increase. The SHBG is known for its U-shaped trajectory in women during the decades of life. An all-time low has been suggested somewhere between the sixth and seventh decade of life [23,24]. The differences found in SHBG levels in this study might be related to the difference in age between the two groups. Thus, the younger HYST group was still reaching towards the lowest SHBG levels and the older HYST + BSO group was turning upwards in between the follow-ups. However, these changes were non-significant, and more research is needed to understand if this is a repeatable finding. Although some studies have suggested that SHBG levels might be related to other comorbidities such as metabolic syndrome, diabetes, and coronary heart disease (CHD), the mechanisms for these relationships are still largely unknown and a possible field for future investigation. The study also investigated the within-group hormonal changes with an expected general decline of the sex hormone levels with time. Between the short-term and long-term follow-up, there was a notable drop-out number in the HYST group which must be considered when interpreting the results. However, given that the study also reported the within-group hormone changes, this gives some verification of the results by showing an expected path of declination with age in both groups. This indicates that the hormonal results are likely to be representative despite the drop-out numbers. 

Based on previous studies showing an NNT ranging between 273 and 323 for bilateral salpingo-oophorectomy to avoid one case of ovarian cancer and the results of the present study finding a small association between bilateral salpingo-oophorectomy and decreased sexual function, this should be considered when deciding upon concomitant bilateral salpingo-oophorectomy at time of hysterectomy on benign indications. However, more prospective studies on the subject are needed.

This study was a prospective cohort study with the advantages of an examination of multiple variables, sexual function and changes of hormone levels, and the surgical procedure. Reporting a long-term perspective provided an opportunity to study variables over time and thus created greater certainty in how the choice of surgery might affect the outcome for these women. With a prospective design, the risk of selection bias was minimized. To date, few other prospective long-term follow-up studies are to be found.

One limitation of the study was the number of dropouts during the long-term follow-up period and warrants consideration when interpreting the results. The study was a patient preference-selection study and not a randomized trial. This can result in heterogenicity between groups and therefore might have affected the results. Further, the exclusion criteria for taking part in the study were rather strict. Psychiatric disorders such as minor depression treated with pharmaceuticals led to exclusion. This might influence the study’s representativity of the general population This study did not include sufficient data of HRT treatment of the included women and the interpretation of the results must take this into consideration. There where data on self-reported HRT, but after cross referencing these data with self-reported drug treatments, a large quantity of the cases seemed to have confused different types of treatments; therefore, the decision was taken to exclude this from the final analyses as there were questions regarding the validity. Caution is also relevant when considering that the loss of patients to the follow-up investigation made it impossible to study changes on patient level over time. The small number of women that left hormonal samples on both the short- and long-term follow up were too few to enable direct comparisons. This, together with the small number of women in the HYST + BSO group that completed the long-term follow-up, warrants caution when interpreting the results.

For further studies of sexual function after hysterectomy with or without concomitant bilateral salpingo-oophorectomy, it would also be of interest to study potential comorbidities such as cardiovascular morbidity, CHD and metabolic syndrome. Data of HRT treatment and a greater and possibly age-controlled study material including participants with other backgrounds than the Caucasian background could minister further interpretation and conclusion. 

In conclusion, the study results, with 10–12 years of follow-up time, found the long-term effect on specific aspects of sexual function to be worsening in women undergoing hysterectomy with concomitant bilateral salpingo-oophorectomy compared to hysterectomy alone. This would favor hysterectomy over hysterectomy with concomitant bilateral salpingo-oophorectomy for preservation of sexual function when conducting HYST on benign indication.

## Figures and Tables

**Table 1 jcm-12-04976-t001:** Exposure characteristics. Mean values with standard deviations (SD), results from *t*-test.

	HYST (n = 115)	HYST + BSO (n = 70)	*p*-Value
Mean age at follow-up (SD)	62 (3)	65 (2.7)	<0.01
Mean weight in kg (SD)	72 (13.4)	71 (11.1)	0.68
Mean age at menarche in years (SD)	13 (1.4)	13 (1.3)	0.76
Mean number of pregnancies (SD)	3 (1.2)	2 (1.2)	0.19
Mean number of deliveries (SD)	2 (1)	2 (1)	0.31

**Table 2 jcm-12-04976-t002:** Sex hormone levels for the HYST and HYST + BSO groups at short- and long-term follow-up, results from *t*-test with standard deviations (SDs).

**Testosterone (nmol/L)**	**HYST**	**HYST + BSO**	***p*-Value**
One-year mean (SD) (n)	1.64 (1.29) (110)	1.48 (0.56) (23)	0.35
Long-term mean (SD) (n)	1.03 (0.75) (62)	0.85 (0.53) (25)	0.20

**Estradiol (nmol/L)**	**HYST**	**HYST + BSO**	***p*-Value**
One-year mean (SD) (n)	0.36 (0.31) (113)	0.29 (0.18) (21)	0.17
Long-term mean (SD) (n)	0.14 (0.18) (61)	0.10 (0.09) (26)	0.17

**SHBG ^A^ (nmol/L)**	**HYST**	**HYST + BSO**	***p*-Value**
One-year mean (SD) (n)	73.81 (42.97 (112)	60.09 (23.06) (21)	0.04
Long-term mean (SD) (n)	68.29 (30.38) (53)	68.22 (27.20) (24)	0.99

**FAI ^B^**	**HYST only**	**HYST + BSO**	***p*-Value**
One-year mean (SD) (n)	3.25 (5.95) (109)	2.70 (1.06) (21)	0.38
Long-term mean (SD) (n)	1.68 (1.30) (51)	1.28 (1.12) (22)	0.18

^A^: Sex-hormone-binding globulin, ^B^: Free androgen index.

**Table 3 jcm-12-04976-t003:** Rank–mean for HYST group in comparison with HYST + BSO group on each of the items in the McCoy sexual functioning questionnaire, results of Mann–Whitney U test.

	HYST	HYST + BSO	
Mccoy Item	Rank–Mean	Rank–Mean	*p*-Value
Satisfaction with present sexual activity level (n)	85.54 (108)	88.12 (64)	0.74
Frequency of sexual fantasies during last month (n)	94.09 (108)	76.72 (66)	0.02
Enjoyment of sexual activity (n)	70.16 (85)	56.70 (45)	0.05
Frequency of feeling arousal during sexual activity (n)	68.31 (80)	53.56 (45)	0.03
Orgasmic frequency (n)	69.87 (82)	49.91 (43)	0.01
Frequency of vaginal dryness (n)	61.72 (83)	70.91 (46)	0.18
Frequency of dyspareunia (n)	60.43 (82)	70.50 (45)	0.13
Satisfaction with the partner as lover (n)	64.88 (84)	66,63 (46)	0.80
Satisfaction with partner as friend (n)	73.76 (92)	73.20 (54)	0.40
Prior coital frequency before gynecological condition (n)	71.27 (96)	77.06 (53)	0.41

## Data Availability

The data presented in this study can be made available on request from the corresponding author. The data are not publicly available due to Swedish laws on research ethics and individual integrity. Before data can be made available, necessary applications to the Swedish Ethical Review Authority and Regin of Västra götaland, Sweden.

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
