# Peer review of "Long-Term Follow Up of Sexual Function and Steroid Levels in Women after Perimenopausal Hysterectomy with or without Concomitant Oophorectomy"

_jcm, 2023, doi:10.3390/jcm12154976_

Round 1

Reviewer 1 Report (New Reviewer)

Thank you for the opportunity to review this manuscript. Sexual function is an important outcome that warrants more research. These findings could impact patient counseling and surgical decision  making.

Suggested changes:

Remove "The function of the uterus is to harbour a pregnancy". Instead, I suggest rephrasing to "The uterus does not exert any endocrine function."  We know some patients attach value to their uterus beyond it's ability to host a pregnancy. Additionally, this sentence is not necessary to make your point.

Line 47: Remove "past their reproductive years". This is confusing because people often refer to a certain age range as "reproductive years" even for patients who are s/p permanent contraception, not sexually active, or has a same-sex partner. Also, some patients may be presented the option of a hysterectomy for medical reasons (severe pelvic pain, dangerous bleeding) even when they would consider additional pregnancies in the future if they didn't have that medical condition. I think the word "advised" in also too strong in this sentence. Consider rephrasing to something like "Surgical removal of the uterus is a reasonable definitive option for women experiencing benign uterine diseases impacting quality of life or resulting in chronic problems." 

Line 53: typo? "and along with properly substitution with estrogens"

Line 76: Rephrase. "This patient material"? Does that mean the study sample?

Line 86  - Grammar error lines 86-88

Methods:

Line 132: why was a paired t test not performed?

Results: 

Do you include the cohort-specific response rate? That would be important to note. 

Line 136-137 - Rephrase this first sentence. It is unclear

Table 2: typo in first column

Table 3:  Needs to update column headers to clarify which cohort is in which column. Typo in column 3.  Why do the n= vary for each item. Is this because participants skipped multiple items? Some of these items were only completed by less than half of the Hyst-bso cohort.

Given the older age group in the hyst-bso cohort, did you consider comparing cohort outcomes by age groups? 

I recommended a few small revisions to wording/grammar in the text above. 

Author Response

Dear recipient,

Thank you for your comments on our paper.

We have removed "The function of the uterus is to harbour a pregnancy" and rephrased the sentence according to your suggestion. See line 44.

We have adjusted the text according to your suggestion regarding "Surgical removal of the uterus is a reasonable definitive option for women experiencing benign uterine diseases impacting quality of life or resulting in chronic problems.”

Typo corrected regarding estrogen substitution on line 55.

The beginning om line 88 now reads “The study sample…”

The grammatical errors regarding the two types of surgical procedures is corrected, line 104-106.

Methods

The use of independent sample t-test was based on advice from a statistician, the argument was that we were not comparing, for example weight at short- and long-term follow-up but rather the differences between each group at the long-term follow-up.  

Results

The first line in the Results is clarified, and now reads: “The HYST group were on average 3 years younger than the HYST+BSO group (Table 1).” Line 158-160.

The typo in Table 2 is corrected.

Questions regarding Table 3.

The groups in Table 3 is clarified.

The typo in column 3, last line is corrected.

Regarding the variance on n=

Yest, it is related to participants skipping individual questions in the McCoy scale. It is somewhat puzzling, it would have been easier to interpret if the either completed it or not.  We have added some text in the discussion regarding this, speculating around if this may be an indication of some of them being sexually inactive. We have another previous section in the discussion elaborating on the rather narrow definition of sexually active (intercourse in the last 30 days) in the McCoy scale. See line 259-263, and line 251-259.

Regarding the question on consider comparing cohort outcomes by age groups?

Yes, we considered that, but we were not able to produce sensible results given that loss to follow-up. The groups got too small.

Kind regards,

Jonas Hermansson and the Author team

Reviewer 2 Report (Previous Reviewer 2)

See comments below

Did You use 10 or 9 item questionnaire (see lines 26,103, 122 and 229)

Pleaae give the reference that used systemic HRT among qwomen over 567 is not recommended (line 244)

Author Response

Dear recipient,

Thank you for your comments on our paper.

We use the 10-item version of the McCoy scale, we have changed 251 in the revised manuscript.

A recently updated reference is added to support that, from a Swedish perspective, HRT is not recommended to treat menopausal symptoms for older women. See line 270-273.

Kin regards,

Jonas Hermansson and the Author team

This manuscript is a resubmission of an earlier submission. The following is a list of the peer review reports and author responses from that submission.

Round 1

Reviewer 1 Report

The authors aimed to assess the long-term (10-12 years) effects on sexual function and changes in sex hormone levels in women having undergone perimenopausal hysterectomy (HYST), with or without concomitant bilateral salpingo-oophorectomy (BSO). 

12 years ago, they performed a prospective cohort study on the same issue during which 441 women (90%; HYST; n = 271 and HYST+BSO; n = 170) completed a one-year survey after the initial surgery. The selection of a type of surgical procedure was based on individual cancer risk aversion.

The current study includes 185 cases of 441 original group (41.9%) and assesses changes in testosterone, Estradiol, SHGB, FAI, and sexual satisfaction based on McCoy’s sex scale

The authors aim to complement the previous study of Aziz et al 2005.

Thus, I have included the data from the original perspective cohort study below to ask a question about the current work which is not clear enough for the reviewer.

“When the HYST and HYST + BSO groups were compared preoperatively, it was found that the HYST group had significantly higher scores in items satisfaction with present sexual life (p ≤ 0.05), sexual enjoyment (p ≤ 0.05), sexual arousal (p ≤ 0.05), orgasmic frequency (p ≤ 0.5), relationship to partner as a lover (p ≤ 0.5), coital frequency (p ≤ 0.05) and in subscales sexual desire (p ≤ 0.01) and the total score (p ≤ 0.01). 

At 1-year follow-up, the HYST group scored significantly higher than the HYST + BSO group on items satisfaction with present sexual life (p ≤ 0.05), sexual enjoyment (p ≤ 0.05), sexual arousal (p ≤ 0.05), orgasmic frequency (p ≤ 0.05) and relationship to partner as a lover (p ≤ 0.05) and on subscales sexual satisfaction (p ≤ 0.01) and total sexual score (p ≤ 0.01). 

When comparing the preoperative and 1-year scores, the HYST group had lower 1-year scores on three of the 14 sexual parameters (coital frequency (p ≤ 0.01), sexual enjoyment (p ≤ 0.05) and total score (p ≤ 0.05), while no significant differences were found in the HYST + BSO group regarding any item or subscale” [Aziz et al 2005]. 

This "pasted" piece of the previous manuscript indicates that the younger HYST group had higher scores on McCoy’s sex scale preoperatively as well as at 1-year follow-up than the older HYST-BSO group. In 2005 Aziz A. did not find a difference in changes after 1 year either in the HYST group or HYST+BSO. Thus, the conclusion was that both groups did not differ in terms of changes between the preoperative and 1-year postoperative scores (Delta). 

Notably, the difference was significant between the compared groups. 

The question is if the results (sex blood hormones, MFSQ) currently collected from 185 patients have been compared with the results of the same patients treated 12 years ago. If so it could be underlined that we have differences in changes inside the particular group or have differences between two groups. The second variant seems to be obvious (because such differences were found preoperatively and at 1-year follow-up).

Author Response

Comments and Suggestions for Authors – reviewer 1

The authors aimed to assess the long-term (10-12 years) effects on sexual function and changes in sex hormone levels in women having undergone perimenopausal hysterectomy (HYST), with or without concomitant bilateral salpingo-oophorectomy (BSO).

12 years ago, they performed a prospective cohort study on the same issue during which 441 women (90%; HYST; n = 271 and HYST+BSO; n = 170) completed a one-year survey after the initial surgery. The selection of a type of surgical procedure was based on individual cancer risk aversion.

The current study includes 185 cases of 441 original group (41.9%) and assesses changes in testosterone, Estradiol, SHGB, FAI, and sexual satisfaction based on McCoy’s sex scale

The authors aim to complement the previous study of Aziz et al 2005.

Thus, I have included the data from the original perspective cohort study below to ask a question about the current work which is not clear enough for the reviewer.

“When the HYST and HYST + BSO groups were compared preoperatively, it was found that the HYST group had significantly higher scores in items satisfaction with present sexual life (p ≤ 0.05), sexual enjoyment (p ≤ 0.05), sexual arousal (p ≤ 0.05), orgasmic frequency (p ≤ 0.5), relationship to partner as a lover (p ≤ 0.5), coital frequency (p ≤ 0.05) and in subscales sexual desire (p ≤ 0.01) and the total score (p ≤ 0.01).

At 1-year follow-up, the HYST group scored significantly higher than the HYST + BSO group on items satisfaction with present sexual life (p ≤ 0.05), sexual enjoyment (p ≤ 0.05), sexual arousal (p ≤ 0.05), orgasmic frequency (p ≤ 0.05) and relationship to partner as a lover (p ≤ 0.05) and on subscales sexual satisfaction (p ≤ 0.01) and total sexual score (p ≤ 0.01).

When comparing the preoperative and 1-year scores, the HYST group had lower 1-year scores on three of the 14 sexual parameters (coital frequency (p ≤ 0.01), sexual enjoyment (p ≤ 0.05) and total score (p ≤ 0.05), while no significant differences were found in the HYST + BSO group regarding any item or subscale” [Aziz et al 2005].

This "pasted" piece of the previous manuscript indicates that the younger HYST group had higher scores on McCoy’s sex scale preoperatively as well as at 1-year follow-up than the older HYST-BSO group. In 2005 Aziz A. did not find a difference in changes after 1 year either in the HYST group or HYST+BSO. Thus, the conclusion was that both groups did not differ in terms of changes between the preoperative and 1-year postoperative scores (Delta).

Notably, the difference was significant between the compared groups.

The question is if the results (sex blood hormones, MFSQ) currently collected from 185 patients have been compared with the results of the same patients treated 12 years ago. If so it could be underlined that we have differences in changes inside the particular group or have differences between two groups. The second variant seems to be obvious (because such differences were found preoperatively and at 1-year follow-up).

Author reply: That is observant and should be clarified. It is not a direct comparison between the women that provided either blood samples or filled out questionnaires at each follow-up. The comparison is between the respective groups at each time point. This is clarified in the text at line 101 on page 3, and line 296 on page 7.

Reviewer 2 Report

The effect of hysterectomy onsexual life differ from one women to another The usual complaints after hysterectomy include the loss of libido, decrease frequency of intercourse, decreased sexual responsiveness, difficulty with reaching orgasm, diminished sensation of the vagina, dyspareunia (painful intercourse),
vaginal shortening, loss of penile penetration, and loss of vaginal elasticity and lubrication.  Moreover majority  of patients were suffering from feelings of premature aging and loss of libido but these symptoms were  improved considerably in most patients.  On the other hand the long term data on this pivotal issue arelkacking and therefore this paper is very interesting from clinical point of view however in my opinion some items should be clarified.

1.       It would be interesting to know what percentage of patients were still sexually active after 10-12 years. This point should be definitely  included in the article - whether the condition of sexual activity was still valid during follow-up after 10-12 years after the procedure.

2.       In what units the measurements of hormones were done? In the table we can only see values related to SD. When  discussing the results, maybe it would also be worthwhile to include the values of the most important results.

3.       How is it possible to achieve almost identical hormone level results in follow up after one year and after 10-12 years? This would suggest that the ovaries show no change in hormonal activity over the years.  Please  clarify this issue.

Author Response

Comments and Suggestions for Authors – reviewer 2

The effect of hysterectomy on sexual life differs from one woman to another. The usual complaints after hysterectomy include the loss of libido, decrease frequency of intercourse, decreased sexual responsiveness, difficulty with reaching orgasm, diminished sensation of the vagina, dyspareunia (painful intercourse), vaginal shortening, loss of penile penetration, and loss of vaginal elasticity and lubrication. Moreover, majority of patients were suffering from feelings of premature aging and loss of libido, but these symptoms were improved considerably in most patients. On the other hand, the long-term data on this pivotal issue are lacking and therefore this paper is very interesting from clinical point of view however in my opinion some items should be clarified.

  1. It would be interesting to know what percentage of patients were still sexually active after 10-12 years. This point should be definitely included in the article - whether the condition of sexual activity was still valid during follow-up after 10-12 years after the procedure.

Author reply: This is a relevant question, in hindsight, this should have been included in the questionnaire. In the analyses in this paper, the woman that have answered McCoy are included. Theoretically, we could assume that the ones not answering McCoy at all indicate that they are not sexually active but that was true for two patients so that may not provide a valid explanation. But, if we use the incomplete answers to McCoy, most of them have answered the questions regarding satisfactory with current sexual activity/frequency and the question on enjoyment of sexual activity.

In total, 10 women responded with a value of either 1 or 2 indicating unsatisfactory level of current sexual activity, 30 women reported some satisfaction or high satisfaction with current level of sexual activity and 3 women reported that they have no enjoyment of sexual activity. All of these 43 women did not complete the rest of the McCoy questions. It is not unlikely to interpret that these women were not sexually active, if so, the made up 23% of the total sample.

In this study, the shorter version of the McCoy questionary was used. If we would have used the full 19 items questionnaire, we would have had access to the question on frequency of sexual intercourse. However, that concerns the last four weeks and only sexual activity in the form of intercourse. It may not entirely capture relevant aspects of whether a person considers them self as sexually active or not. If we would do this study again, we would probably include a question similar to “Do you consider your self to be sexually active?” with the following alternatives Yes, No, No, but I would like to be, and a possibility of a free text option. By doing so, we believe that we would more accurately capture the possible dimensions of this question for aging women.

This is added in the manuscript in the discussion starting on page 6, line 226.

  1. In what units the measurements of hormones were done? In the table we can only see values related to SD. When discussing the results, maybe it would also be worthwhile to include the values of the most important results.

Author response: Thank you for pointing that out, it is included under 3.1. Assays on page 3 and in the table in the result section.

  1. How is it possible to achieve almost identical hormone level results in follow up after one year and after 10-12 years? This would suggest that the ovaries show no change in hormonal activity over the years. Please clarify this issue.

Author response: Another relevant question. We have clarified the test on this so that we more clearly address the potential selection effect of the fact that we did loose a number of patients in this long-term follow-up. We believe that it is vital to interpret the hormonal results with caution. This is mentioned on page 8, line 296.

Round 2

Reviewer 1 Report

If “It is not a direct comparison between the women that provided either blood samples or filled out questionnaires at each follow-up” you should change the entire study you should change this.

Please extract 185 cases from the initial population of 441 women assessed in 2005 and compare their questionaries and blood samples taken after 12 years. Please provide new results.

Author Response

Author response: We have done an additional version of table 2 using only the patients in each group that we have hormonal data on from both short and long-term follow-up. When we did that, the number of patients in the HYST+BSO group was as low as six patients. Therefore, we have completed the results and the discussion with additional text motivating using the comparison between groups, rather than using data from patients that completed both hormonal samples. See line 153-156 page 4. Also see additional table not in the manyscript.

Reviewer 2 Report

I still do not understand why the women without BSO had the same estradiol levels as women after BSO especially in short term observation. Probably These group used Hormone replacement therapy??? Please clarify.

Author Response

Author response: We agree that a variance in HRT may very well hold part in the explanation behind the results on hormonal levels. We have clarified this further in the discussion, line 293-298 page 8. In there, we elaborate on the issues we found when cross referencing the data on self-reported HRT and therefor decided on not including it in the final analyses.

We have done an additional version of table 2 using only the patients in each group that we have hormonal data on from both short and long-term follow-up. When we did that, the number of patients in the HYST+BSO group was as low as six patients. Therefore, we have completed the results and the discussion with additional text motivating using the comparison between groups, rather that using data from patients that completed both hormonal samples. See line 153-156 page 4.
